# PTPRN Serves as a Prognostic Biomarker and Correlated with Immune Infiltrates in Low Grade Glioma

**DOI:** 10.3390/brainsci12060763

**Published:** 2022-06-10

**Authors:** Peng Li, Fanfan Chen, Chen Yao, Kezhou Zhu, Bei Zhang, Zelong Zheng

**Affiliations:** 1VCU Massey Cancer Center, Department of Human and Molecular Genetics, Institute of Molecular Medicine, School of Medicine, Virginia Commonwealth University, Richmond, VA 23298, USA; peng.li@vcuhealth.org (P.L.); kezhou.zhu@vcuhealth.org (K.Z.); 2Neurosurgical Department, Shenzhen Second People’s Hospital, The First Affiliated Hospital of Shenzhen University, Shenzhen 518035, China; cff_126com@126.com (F.C.); yaochen_1987@126.com (C.Y.); 3Department of Biostatistics, Virginia Commonwealth University, Richmond, VA 23298, USA; zhangb11@vcu.edu; 4Department of Neurosurgery, Guangzhou First People’s Hospital, School of Medicine, South China University of Technology, Guangzhou 510180, China

**Keywords:** PTPRN, glioma, immune infiltration, biomarker, tumor microenvironment

## Abstract

Background: Glioma is one of the most common malignant tumors of the central nervous system. Immune infiltration of tumor microenvironment was associated with overall survival in low grade glioma (LGG). However, effects of Tyrosine phosphatase receptor type N (PTPRN) on the progress of LGG and its correlation with tumor infiltration are unclear. Methods: Here, datasets of LGG were from The Cancer Genome Atlas (TCGA) and normal samples were from GTEx dataset. Gepia website and Human Protein Atlas (HPA) Database were used to analyze the mRNA and protein expression of PTPRN. We evaluated the influence of PTPRN on survival of LGG patients. MethSurv was used to explore the expression and prognostic patterns of single CpG methylation of PTPRN gene in LGG. The correlations between the clinical information and PTPRN expression were analyzed using logistic regression and Multivariate Cox regression. We also explored the correlation between PTPRN expression and cancer immune infiltration by TIMER. Gene set enrichment analysis (GSEA) was formed using TCGA RNA-seq datasets. Results: PTPRN mRNA and protein expression decreased in LGG compared to normal brain tissue in TCGA and HPA database. Kaplan-Meier analysis showed that the high expression level of PTPRN correlated with a good overall survival (OS) of patients with LGG. The Multivariate Cox analysis demonstrated that PTPRN expression and other clinical-pathological factors (age, WHO grade, IDH status, and primary therapy outcome) significantly correlated with OS of LGG patients. The DNA methylation pattern of PTPRN with significant prognostic value were confirmed, including cg00672332, cg06971096, cg01382864, cg03970036, cg10140638, cg16166796, cg03545227, and cg25569248. Interestingly, PTPRN expression level significantly negatively correlated with infiltrating level of B cell, CD4+ T cells, Macrophages, Neutrophils, and DCs in LGG. Finally, GSEA showed that signaling pathways, mainly associated with tumor microenvironment and immune cells, were significantly enriched in PTPRN high expression. Conclusion: PTPRN is a potential biomarker and correlates with tumor immune infiltration in LGG.

## 1. Introduction

Glioma is one of the most common malignant tumors of the central nervous system. According to the glioma grading classification criteria of the World Health Organization (WHO), gliomas are divided into Grade 1, 2, 3, and 4 [1,2]. Grade 1 and 2 gliomas belong to low-grade gliomas (LGGs) while Grade 3 and 4 gliomas are high-grade gliomas (HGGs). LGG encompasses a diverse group of diffusely infiltrative, slowly growing glial brain tumors that would dedifferentiate and progress to HGG. Standard treatment strategies for LGG include surgical treatment, radiotherapy, chemotherapy, and combination therapy. Despite advances in surgical techniques and the availability of new chemotherapeutic agents, outcomes for patients with LGG remain poor.

Cancer immunotherapy has recently become an important pillar of cancer treatment. Immunotherapy based on cytotoxic T lymphocyte-associated antigen 4 (CTLA4), programmed death-1 (PD-1) and programmed death ligand-1 (PD-L1) inhibitors have emerged as an effective treatment in melanoma, non-small cell lung carcinoma and glioma [3,4,5]. Tumor-infiltrating lymphocytes, such as tumor-associated macrophages (TAMs), play a very important role in patient prognosis and the efficacy of immunotherapy [6]. However, the discovery of glioma immunotherapy mainly affects GBM and rarely appears in LGG research. Additionally, in recent years, with the introduction of the concept of precision medicine, the detection of molecular diseases and the improvement of targeted therapy have significantly improved the overall survival rate of LGG patients. In patients with LGG, 1p/19q co-deletion, IDH mutation, and TERTp (telomerase reverse transcriptase gene promoter region) mutation can help assess the prognosis [7,8]. LGG patients with IDH1 mutation or 1p/19q co-deletion but not TERTp mutation can benefit from radiotherapy and chemotherapy. Although molecular targeted therapy has shown good clinical effects, the curing of LGG patients is still a challenge, especially due to the development of drug resistance. Therefore, there is an urgent need to develop a new and effective biomarker for the diagnosis and treatment of patients with LGG.

PTPRN, also called islet cell autoantigen 512 (ICA512/IA-2), is located on human chromosome band 2q35. PTPRN is mainly expressed in endocrine cells, neurons of the autonomic nervous system, and neuroendocrine neurons of the brain, including pancreas, pituitary, adrenal medulla, amygdala, and hypothalamus, because they all contain neuro-secretory granules [9]. PTPRN is a type I transmembrane protein that participates in neuroendocrine processes, such as biogenesis, transport, and/or regulation of exocytosis [10]. It’s involved in regulating the secretion pathways of various neuroendocrine cells and in the occurrence and development of diabetes mellitus [11]. Multiple studies have concluded that PTPRN may play a potential role in solid tumors [12,13,14,15,16]. PTPRN was identified as an independent prognostic factor in hepatocellular carcinoma [16]. Abnormal hypermethylation of PPTRN DNA is related to the OS of patients with ovarian cancer [12]. In addition, the expression of PTPRN mRNA and protein increased in breast cancer cells under hypoxia [13]. PTPRN was overexpressed in a group of glioblastoma patients, indicating poor survival [14]. Furthermore, PTPRN could be a biomarker that predicts the prognosis of glioblastoma and could estimate the OS of patients with glioblastoma [15]. However, the role of PTPRN in LGG is still unknown, especially its role in the regulation of tumor microenvironments.

In this study, the clinical features and survival information of patients with LGG from TCGA was analyzed using bioinformatics to assess the prognostic significance of PTPRN in LGG. We also investigate the relationship between PTPRN expression and tumor-infiltrating immune cells in LGG. Finally, the functional enrichment analysis was performed based on the differential genes expression in low and high PTPRN expression in LGG. These results shed light on the essential role of PTPRN and provide a potential mechanism for LGG.

## 2. Material and Methods

### 2.1. PTPRN Expression and Overall Survival Analysis by Gepia

Using an open website, Gepia (http://gepia2.cancer-pku.cn/#analysis, accessed on 9 May 2022), to analyze the mRNA expression of PTPRN between normal patient samples and LGG samples. All the samples are from TCGA (*n* = 518) and GTEx (*n* = 207) database. Fragments per kilobase million (FPKM) values were transformed into per kilobase million (TPM) values, which was more comparable between samples. A P value less than 0.05 was considered significant by Wilcoxon rank sum test. The samples of LGG were divided into two groups according to the median PTPRN expression to plot a Kaplan-Meier survival curve of LGG patients (*n* = 257 in both high and low expression groups, two cases were missing during follow-up).

### 2.2. The Human Protein Atlas Database

The human protein atlas (https://www.proteinatlas.org, accessed on 17 February 2022) is a Swedish-based program to map all the human proteins in cells, tissues, and organs using an integration of various omics technologies, including, antibody-based imaging, mass spectrometry-based proteomics, transcriptomics, and systems biology. It was used to compare the protein expression of PTPRN in LGG and normal brain tissue.

### 2.3. Clinic Data Analysis

We extracted the clinic data of the LGG project from TCGA (https://portal.gdc.cancer.gov/, accessed on 1 February 2022) [17]. The correlations between PTPRN expression and clinic-pathological parameters (histological type, IDH status, 1p/19q codeletion, and WHO grade) in LGG patients were analyzed [18].

### 2.4. DNA Methylation of the PTPRN Gene

DNA methylation plays an important function in prognostic evaluation and potential biomarker in tumorigenesis and progression. We used an integrated online tool, MethSurv (https://biit.cs.ut.ee/methsurv/, accessed on 2 May 2022), to explore the expression and prognostic patterns of single CpG methylation of PTPRN gene in LGG [19,20,21]. The DNA methylation results of PTPRN and survival analysis were generated via the MethSurv platform.

### 2.5. TIMER Database to Analyze Tumor Infiltrating Immune Cells

The TIMER database (https://cistrome.shinyapps.io/timer/, accessed on 5 February 2022), which includes 10,897 samples across 32 cancer types from TCGA, is a comprehensive resource for estimating the abundance of immune infiltrates cells, including B cells, CD4+ T cells, CD8+ T cells, neutrophils, macrophages, and DCs [22]. We analyzed the correlation of PTPRN expression with the abundance of immune infiltrates via the gene module. For each survival analysis, TIMER outputs Kaplan-Meier plots for TIICs and genes to visualize the survival differences between the upper and lower 50 percentile of patients. A log-rank *p*-value is also calculated and displayed for each Kaplan-Meier plot.

### 2.6. Gene Set Enrichment Analysis

A computational method that analyzes the statistical significance of a priori defined set of genes and the existence of concordant differences between two biological states is known as the GSEA [23]. In this study, GSEA created an initial list on the classification of the genes according to their correlation with the PTPRN expression. This computational method expounded on the noteworthy differences that we observed in the survival between high- and low- PTPRN expression groups. For each analysis, we performed 1000 repetitions of gene set permutations. The phenotype label that we put forth was the expression level of PTPRN. Additionally, to sort the enriched pathways in each phenotype, we utilized the nominal *p*-value and normalized enrichment score (NES). Gene sets with a discovery rate (FDR) <0.050 were considered to be significantly enriched.

### 2.7. Statistical Analysis

All *p*-values were two-sided, and values lower than 0.050 were considered significant. The correlations between the clinical information and PTPRN expression were analyzed using logistic regression. Multivariate Cox analysis was used to evaluate the influence of PTPRN expression and other clinic-pathological factors on survival. The CIBERSORT package was used to explore the difference in immune cells subtypes. 

## 3. Results

### 3.1. PTPRN Expression Levels and Prognostic Value in LGG 

We used Gepia website to determine the expression difference of PTPRN mRNA level between LGG and normal brain tissues. Compared with that in normal tissues, PTPRN mRNA expression was significantly decreased (*p* < 0.050) (Figure 1A) in LGG. Meanwhile, the protein of PTPRN expression was consistent with the mRNA results (Figure 1B). Kaplan-Meier analysis showed that high expression levels of PTPRN correlated with a good overall survival (OS) of patients with glioma (*p* = 0.000) (Figure 1C). 

Then, we explored the associations between PTPRN expression and clinical characteristics, such as histological type, IDH status, 1p/19q status, and grade. In oligodendroglioma, PTPRN expression was sharply higher than astrocytoma (*p* = 0.000) (Figure 2A). Additionally, PTPRN expression was significantly higher in IDH mutation status (*p* = 0.000) (Figure 2B) and 1p/19q codeletion status gliomas (*p* = 0.000) (Figure 2C). However, the expression of PTPRN was similar in different grades of glioma (*p* = 0.101) (Figure 2D). Moreover, we examined whether PTPRN expression was an independent prognostic factor for LGG using Multivariate Cox regression analyses. The Multivariate Cox analysis demonstrated that age (*p* = 0.000), WHO grade (*p* = 0.000), IDH status (*p* = 0.001), primary therapy outcome (*p* = 0.000), and PTPRN expression (*p* = 0.016) significantly correlated with OS of LGG patients (Table 1). These results indicate that PTPRN expression is an independent prognostic index and high expression of PTPRN is correlated with longer OS.

### 3.2. DNA Mythylation Analysis of the PTPRN Gene in LGG

DNA methylation is an epigenetic alteration that relates to the tumorigenesis and progression. DNA methyltransferases on CpG island methylation are transcription factors that can suppress or promote cell growth and the process is a reversible. We showed the heatmap of DNA methylation clustering the expression levels of the PTPRN gene in LGG (Figure 3A). Furthermore, the DNA methylation pattern of PTPRN with significant prognostic value were also confirmed, such as cg00672332, cg06971096, cg01382864, cg03970036, cg10140638, cg16166796, cg03545227, and cg25569248 (Figure 3B–I, and Table 2).

### 3.3. Correlation of PTPRN Expression with Immune Infiltration Level and Cumulative Survival in LGG

As mentioned above, some tumor-infiltrating lymphocytes are independent predictors of cancer survival, and thus, we investigated the association of PTPRN expression and immune infiltration levels in LGG. We implemented it by selecting PTPRN expression levels that were positively correlated with tumor purity. The investigation showed that the level of PTPRN expression negatively correlated with the infiltration level of B cell (*r* = −0.370, *p* = 0.000), CD4+ T cells (*r* = −0.518, *p* = 0.000), Macrophages (*r* = −0.485, *p* = 0.000), Neutrophils (*r* = 0.324, *p* = 0.000) and DCs (*r* = −0.403, *p* = 0.000) in LGG (Figure 4A). Moreover, our findings showed that B cell (*p* = 0.000), CD8+ T cells (*p* = 0.010), CD4+ T cells (*p* = 0.000), Macrophages (*p* = 0.000), Neutrophils (*p* = 0.000) and DCs (*p* = 0.001) are factors related to the cumulative survival rate of LGG over time (Figure 4B). These data strongly indicate that PTPRN is associated with immune infiltration in LGG.

### 3.4. Gene Sets Enriched in PTPRN Expression Phenotype

Based on GSEA, PTPRN-associated signaling pathways were used to determine in LGG between low and high expression data sets and demonstrated significant differences (FDR < 0.050, *p*-value < 0.050) in the enrichment of GO and KEGG collection. 

Ten pathways, related to the tumor microenvironment, including interferon signaling, immunoregulatory interactions between lymphoid and lymphoid cell, complement cascade, FCGR3A mediated IL10 synthesis, chemokine receptors bind chemokine, IL3, IL-5 and GM-CSF signaling, type II interferon signaling, Toll-like receptor signaling, NKT pathway, and TNFs bind their physiological receptors were showed significantly differential enrichment in PTPRN high and low expression groups based on NES, FDR, and *p*-value (Figure 5, Table 3). Strikingly, ten pathways, associated with immune cells, such as B cell receptor, TCR signaling, interactions between immune cells and microRNAs in the tumor microenvironment, T helper pathway, TCRA pathway, DC pathway, CTL pathway, B lymphocyte pathway, ASB cell pathway, and granulocytes pathway were demonstrated significantly differential enrichment between PTPRN high and low expression groups according to NES, FDR, and *p*-value (Figure 6, Table 4). 

## 4. Discussion

PTPRN is a transmembrane protein and is expressed in different tumors, including breast cancer, hepatocellular carcinoma, ovarian cancer, and glioblastoma. However, little is known about the potential prognostic impact of PTPRN in LGG. In this investigation, we designed the in silico experiments to determine the clinical value of PTPRN in LGG. 

Here, we used TCGA LGG data to analyze the expression level of PTPRN in both tumor and normal tissues. The result showed that mRNA and protein expression of PTPRN was sharply lower in LGG and LGG patients with high expression of PTPRN had a positive prognosis, which suggests LGG has prognostic value. Moreover, Multivariate COX regression analysis confirmed that high expression of PTPRN was an independent prognostic factor in patients with LGG. Previous studies have shown that the type of astrocytic tumor (vs. oligodendroglioma) is a poor prognostic parameter for LGG patients. In our study, we found that PTPRN expression was significantly lower in astrocytoma than in oligodendrocytoma, which again confirmed the prognostic value of PTPRN expression. Additionally, the relationship between PTPRN expression and IDH status or 1p/19q status was illustrated. The presence of IDH mutations is a powerful prognostic biomarker in patients with glioma and is associated with favorable outcomes independent of age and grade. Therefore, WHO strongly recommends that IDH phenotype is used as a new clinical diagnostic method. Complete deletion of both 1p and 19q (1p/19q co-deletion) is the molecular genetic signature of oligodendrogliomas. The presence of 1p/19q co-deletion is a strong independent prognostic biomarker associated with improved survival in both diffuse low-grade and anaplastic tumors. Our study demonstrated PTPRN expression was higher in LGG patients with IDH mutation and 1p/19q co-deletion. Additionally, DNA methylation plays an important role in prognostic evaluation and potential biomarker in tumorigenesis. the DNA methylation pattern of PTPRN with significant prognostic value were also confirmed, such as cg00672332, cg06971096, cg01382864, cg03970036, cg10140638, cg16166796, cg03545227, and cg25569248. In brief, these results imply that PTPRN could be a promising prognostic factor in LGG.

Immune infiltration in LGG is a hot topic at present. Understanding immune infiltrating cells is conducive to the development of immunotherapy for LGG. A significant association between PTPRN expression and immune infiltration levels in LGG was established. PTPRN expression level significantly negatively correlated with infiltrating level of B cell, CD4+ T cells, Macrophages, Neutrophils, and DCs in LGG. In addition, the relationship between PTPRN and immune cells implicates PTPRN plays a crucial part in regulating the immune microenvironment of LGG. To further investigate the function of PTPRN in LGG, we performed GSEA using TCGA dataset. GSEA showed that signaling pathways, mainly associated with microenvironment and immune cells, were significantly enriched in PTPRN high expression. Our overall finding emphasized the significant effect of PTPRN in immune infiltration in LGG.

PTPRN, termed ICA512 or ICA3 as well, was reported to play a role in vesicle-mediated secretory processes in hippocampus, pituitary, and pancreatic islets and regulate catalytic active protein-tyrosine phosphatases [24,25,26]. Interestingly, PTPRN is highly expressed in patients with high-grade glioma, which activates the PI3K/AKT pathway by interacting with HSP90AA1 to promote cell proliferation and metastasis [10]. However, a Kaplan-Meier survival analysis indicated that expression of PTPRN was downregulated in GBM tissue when compared with that of normal tissue and this gene was a good prognostic biomarker for GBM [27]. Although there is no detailed explanation for these controversial results except the sample numbers analyzed are small, we have reason to believe that PTPRN plays the important role in the development of pathophysiology in LGG.

There are several limitations in this study. First, the findings showed in this study need to be further verified by bench experiments. Second, the current findings only analyzed noticeable features associated with the prognosis of PTPRN without mechanism exploration. Therefore, additional research in vitro and in vivo is needed to confirm PTPRN efficacy as a viable target in the glioma immune microenvironment and to develop glioma immunotherapy in the future.

In summary, our findings demonstrated that high PTPRN expression correlates with good prognosis and restraints immune infiltration levels (in B cell, CD4+ T cell, macrophage, neutrophil, and dendritic cell) in LGG. In addition, there was a large degree of tumor immune cell infiltration in LGG patients with high PTPRN expression. In addition, the key pathways in LGG that were regulated by PTPRN are possibly tumor microenvironment and immune cells. These findings suggest that PTPRN could be an independent prognostic factor and correlates with tumor immune infiltration in LGG.

## 5. Conclusions

PTPRN could be an independent prognostic factor and correlates with tumor immune infiltration in LGG.

## Figures and Tables

**Figure 1 brainsci-12-00763-f001:**
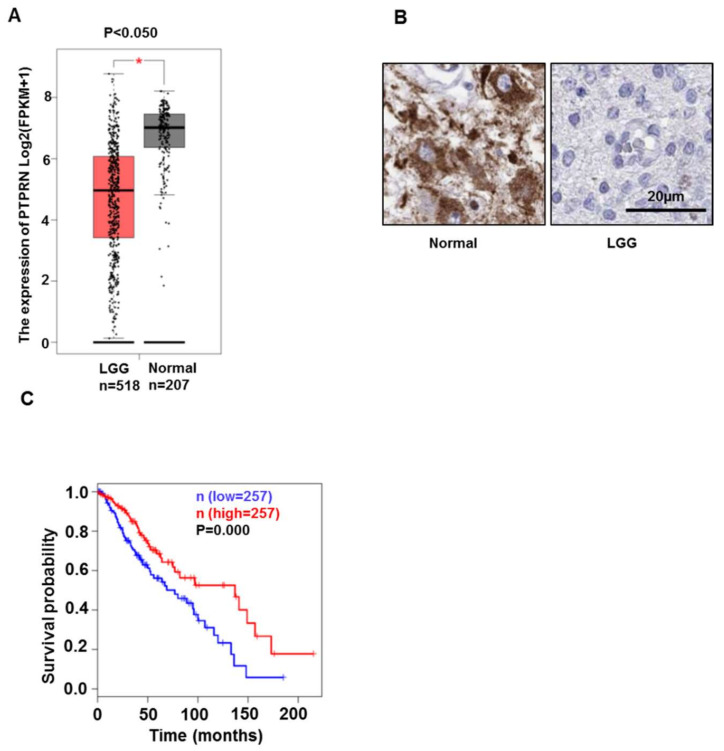
PTPRN mRNA and protein expression between normal and LGG samples, and overall survival in LGG. (**A**) PTPRN mRNA expression is higher in normal brain (*n* = 207) than that in LGG (*n* = 518) from the Gepia website (http://gepia2.cancer-pku.cn/#analysis, accessed on 9 May 2022). mean ± SD, *: *p* < 0.050. (**B**) Representative IHC images of PTPRN protein expression in normal brain tissues and glioma tissue (https://www.proteinatlas.org, accessed on 17 February 2022). (**C**) Overall patient survival in groups of high (red) and low (blue) expression was analyzed by Kaplan-Meier survival curve.

**Figure 2 brainsci-12-00763-f002:**
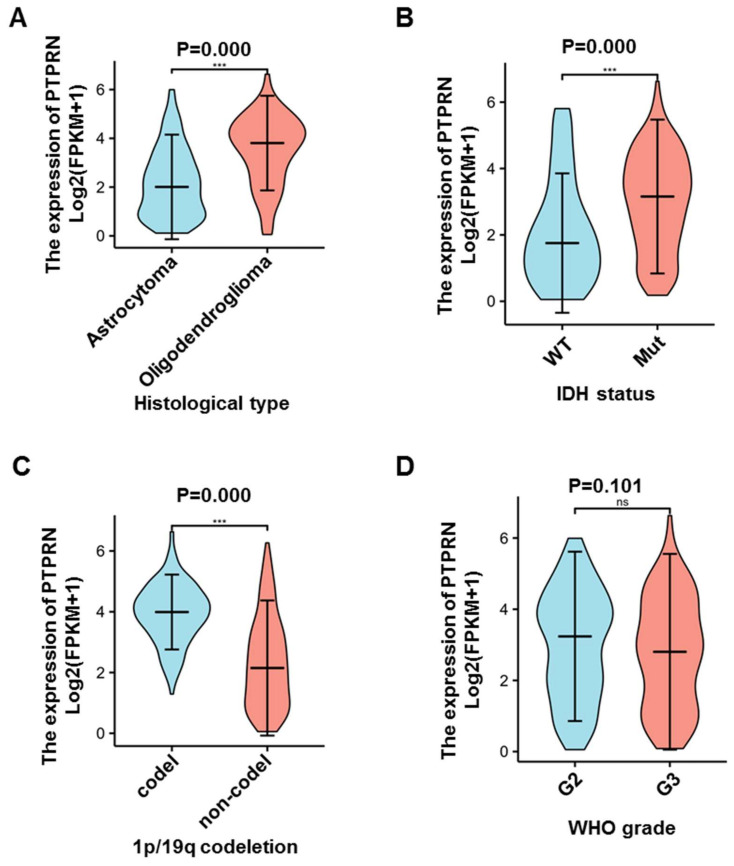
The associations between PTPRN mRNA expression and clinic characteristics. (**A**) In oligodendroglioma, PTPRN expression was higher than astrocytoma (*p* = 0.000). *n* (Astrocytoma) = 195, *n* (Oligodendroglioma) = 199. (**B**) PTPRN expression was high in IDH mutation status (*p* = 0.000) compare to IDH wildtype. *n* (WT) = 95, *n* (Mut) = 428. (**C**) PTPRN expression was elevated in 1p/19q codeletion status gliomas compare to Non-codel status (*p* = 0.000). *n* (Codel) = 171, *n* (Non-codel) = 357. (**D**) The expression of PTPRN was similar in different grades of glioma (*p* = 0.101). *n* (G2) = 224, *n* (G3) = 243. ns: not significant; ***, *p* < 0.001. abbreviation: WT: wildtype; Mut: mutation; Codel: codeletion; Non-codel: non-codeletion.

**Figure 3 brainsci-12-00763-f003:**
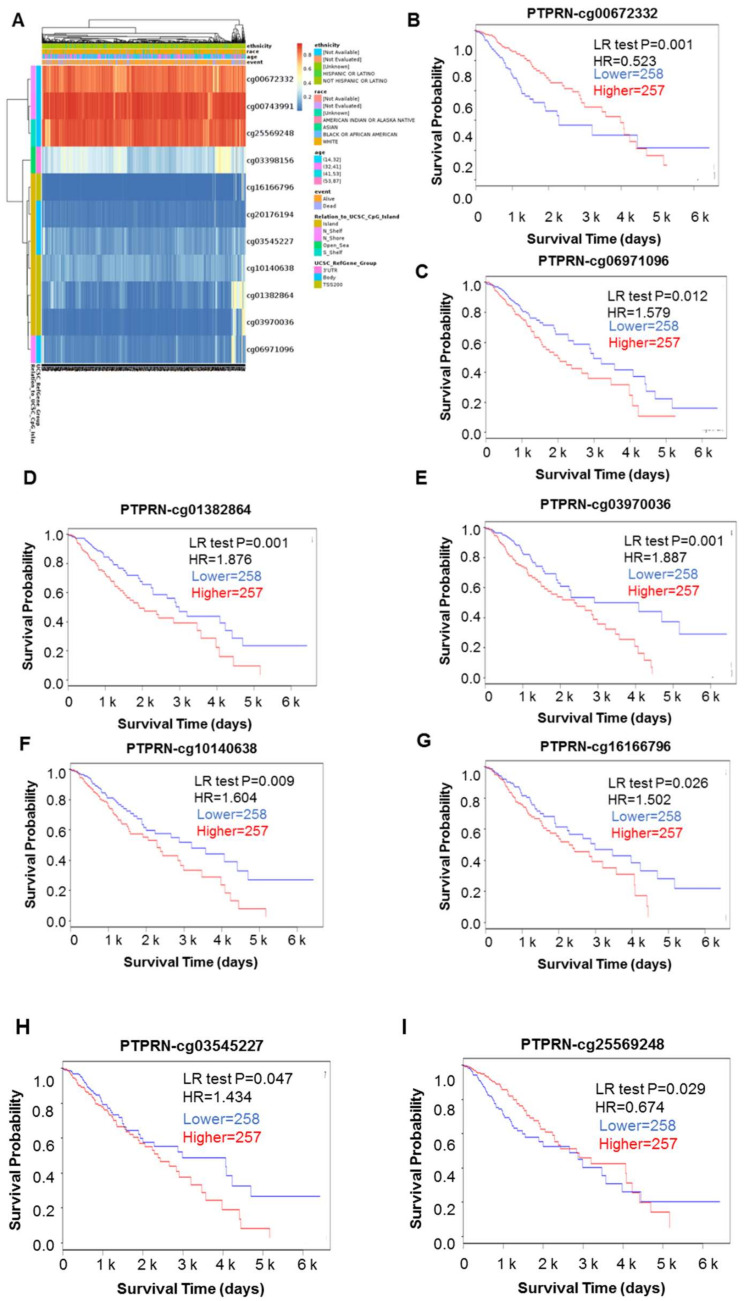
The DNA methylation of PTPRN in LGG of TCGA. (**A**) Heatmap of DNA methylation expression levels of the PTPRN gene in LGG by MethSurv platform. (**B**–**I**) Prognostic values of single CpG of the PTPRN gene in LGG. The threshold of significance was LR Test *p* value <0.05. cg00672332, cg06971096, cg01382864, cg03970036, cg10140638, cg16166796, cg03545227, and cg25569248 of PTPRN displays the significant level of DNA methylation in LGG.

**Figure 4 brainsci-12-00763-f004:**
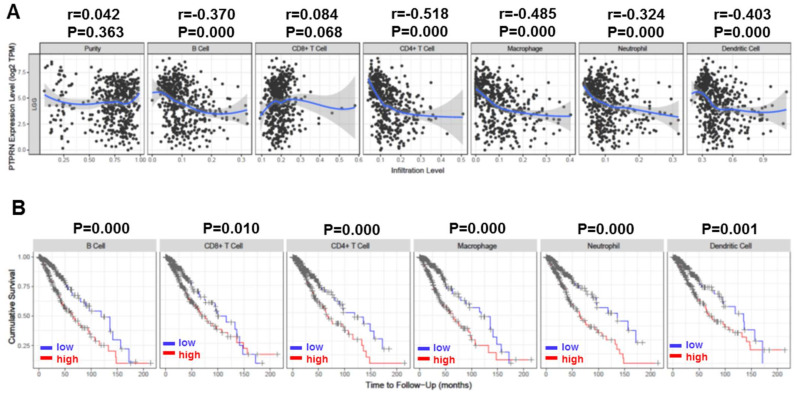
Correlation of PTPRN expression with immune infiltration level in LGG. (**A**) PTPRN expression level has significant negative correlations with infiltrating levels of B cell (*r* = −0.370, *p* = 0.000), CD4+ T cells (*r* = −0.518, *p* = 0.000), Macrophages (*r* = −0.485, *p* = 0.000), Neutrophils (*r* = −0.324, *p* = 0.000) and DCs (*r* = −0.403, *p* = 0.001) in LGG. (**B**) Cumulative survival is related to B cell (*p* = 0.000), CD8+ T cells (*p* = 0.010), CD4+ T cells (*p* = 0.000), Macrophages (*p* = 0.000), Neutrophils (*p* = 0.000) and DCs (*p* = 0.001) in LGG.

**Figure 5 brainsci-12-00763-f005:**
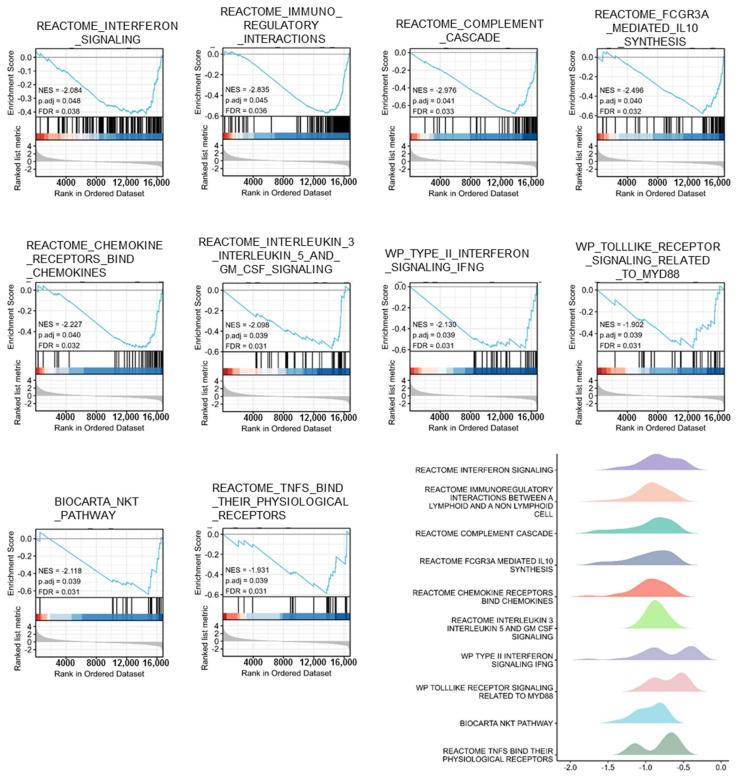
Pathways, related to the microenvironment, including interferon signaling, immunoregulatory interactions between lymphoid and lymphoid cell, complement cascade, FCGR3A mediated IL10 synthesis, chemokine receptors bind chemokine, IL3 IL-5 and GM-CSF signaling, type II interferon signaling, Toll-like receptor signaling, NKT pathway, and TNFs bind their physiological receptors were showed significantly differential enrichment in PTPRN high and low expression groups based on NES, FDR, and *p*-value.

**Figure 6 brainsci-12-00763-f006:**
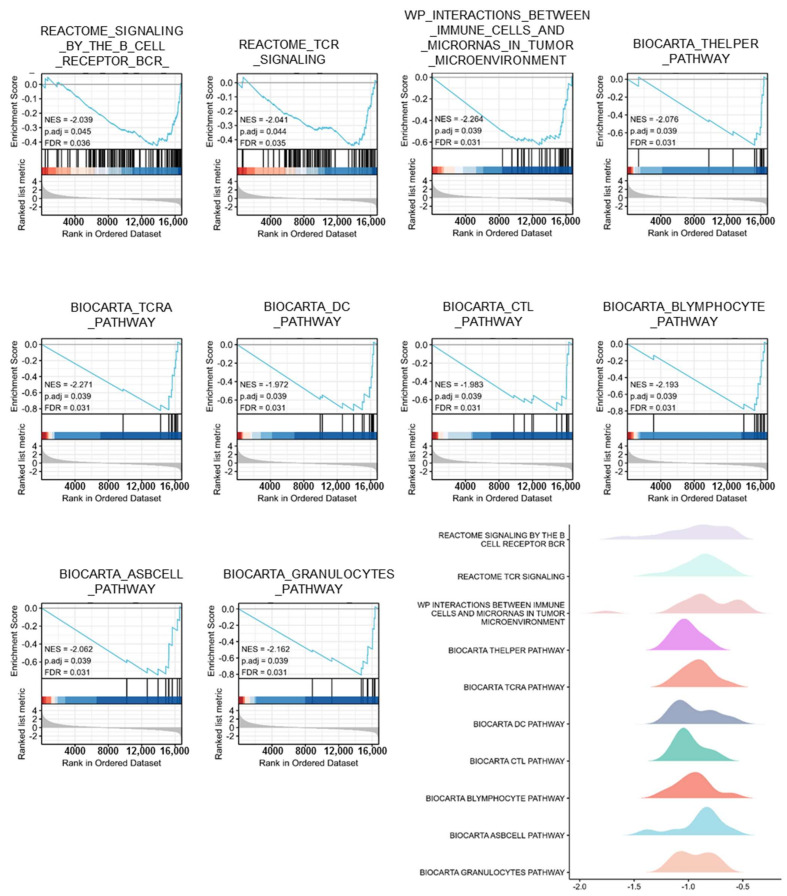
Pathways, associated with immune cells, such as B cell receptor, TCR signaling, interactions between immune cells and microRNAs in the tumor microenvironment, T helper pathway, TCRA pathway, DC pathway, CTL pathway, B lymphocyte pathway, ASB cell pathway, and granulocytes pathway were demonstrated significantly differential enrichment between PTPRN high and low expression groups according to NES, FDR, and *p*-value.

**Table 1 brainsci-12-00763-t001:** Multivariate Cox regression analysis of clinicopathological features (including PTPRN expression) with OS in the TCGA datasets.

Characteristics	Total	HR(95% CI)		*p* Value
Age (≤40 vs. >40)	527	3.245 (2.039–5.164)	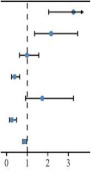	<0.001
WHO grade (G2 vs. G3)	466	2.163 (1.355–3.450)		0.001
Histological type (Astrocytoma vs. Oligodendroglioma)	527	0.988 (0.624–1.564)		0.958
IDH status (WT vs. Mut)	524	0.391 (0.241–0.635)		<0.001
1p/19q codeletion (codel vs. non-codel)	527	1.731 (0.924–3.241)		0.087
Primary therapy outcome (PD&SD vs. PR&CR)	457	0.254 (0.134–0.481)		<0.001
PTPRN (High vs. Low)	527	0.872 (0.781–0.974)		0.016

**Table 2 brainsci-12-00763-t002:** Prognostic Value of Single CpG of the PTPRN gene family in LGG by MethSurv platform.

Gene-CpG	HR	LR Test *p*-Value
PTPRN-Body-N_Shelf-cg00672332	0.523	**0.000 ***
PTPRN-Body-N_Shore-cg00743991	0.817	0.260
PTPRN-Body-N_Shore-cg06971096	1.579	**0.012 ***
PTPRN-TSS200-Island-cg01382864	1.876	**0.001 ***
PTPRN-TSS200-Island-cg03970036	1.887	**0.001 ***
PTPRN-TSS200-Island-cg10140638	1.604	**0.009 ***
PTPRN-TSS200-Island-cg16166796	1.502	**0.026 ***
PTPRN-Body-Island-cg03545227	1.434	**0.047 ***
PTPRN-Body-Island-cg20176194	1.001	1.000
PTPRN-3’UTR-Open_Sea-cg03398156	0.873	0.450
PTPRN-Body-S_Shelf-cg25569248	0.674	**0.029 ***

* : *p* < 0.050.

**Table 3 brainsci-12-00763-t003:** Pathways, related to the microenvironment, were showed significantly differential enrichment in PTPRN high and low expression groups based on NES, FDR, and *p*-value.

ID	NES	*p*. Adjust	FDR
REACTOME_INTERFERON_SIGNALING	−2.084	0.048	0.038
REACTOME_IMMUNOREGULATORY_INTERACTIONS_BETWEEN_A_LYMPHOID_AND_A_NON_LYMPHOID_CELL	−2.835	0.045	0.036
REACTOME_COMPLEMENT_CASCADE	−2.976	0.041	0.033
REACTOME_FCGR3A_MEDIATED_IL10_SYNTHESIS	−2.496	0.04	0.032
REACTOME_CHEMOKINE_RECEPTORS_BIND_CHEMOKINES	−2.227	0.04	0.032
REACTOME_INTERLEUKIN_3_INTERLEUKIN_5_AND_GM_CSF_SIGNALING	−2.098	0.039	0.031
WP_TYPE_II_INTERFERON_SIGNALING_IFNG	−2.13	0.039	0.031
WP_TOLLLIKE_RECEPTOR_SIGNALING_RELATED_TO_MYD88	−1.902	0.039	0.031
BIOCARTA_NKT_PATHWAY	−2.118	0.039	0.031
REACTOME_TNFS_BIND_THEIR_PHYSIOLOGICAL_RECEPTORS	−1.931	0.039	0.031

significance: False discovery rate (FDR) < 0.050 and *p*. adjust < 0.050.

**Table 4 brainsci-12-00763-t004:** Pathways, associated with immune cells, were demonstrated significantly differential enrichment between PTPRN high and low expression groups according to NES, FDR, and *p*-value.

ID	NES	*p*. Adjust	FDR
REACTOME_SIGNALING_BY_THE_B_CELL_RECEPTOR_BCR_	−2.039	0.045	0.036
REACTOME_TCR_SIGNALING	−2.041	0.044	0.035
WP_INTERACTIONS_BETWEEN_IMMUNE_CELLS_AND_MICRORNAS_IN_TUMOR_MICROENVIRONMENT	−2.264	0.039	0.031
BIOCARTA_THELPER_PATHWAY	−2.076	0.039	0.031
BIOCARTA_TCRA_PATHWAY	−2.271	0.039	0.031
BIOCARTA_DC_PATHWAY	−1.972	0.039	0.031
BIOCARTA_CTL_PATHWAY	−1.983	0.039	0.031
BIOCARTA_BLYMPHOCYTE_PATHWAY	−2.193	0.039	0.031
BIOCARTA_ASBCELL_PATHWAY	−2.062	0.039	0.031
BIOCARTA_GRANULOCYTES_PATHWAY	−2.162	0.039	0.031

significance: False discovery rate (FDR) < 0.050 and *p*. adjust < 0.050.

## Data Availability

The data used to support the findings of this study are included with the article.

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
