# Peer review of "PTPRN Serves as a Prognostic Biomarker and Correlated with Immune Infiltrates in Low Grade Glioma"

_brainsci, 2022, doi:10.3390/brainsci12060763_

Round 1

Reviewer 1 Report

Peng Li et al. perform cell deconvolution analysis and correlates the results of this analysis with clinical and transcriptomic data. Their findings suggest that PTPRN is a potential biomarker for tumor immune infiltration in LGG.

1) Graphical results are well presented. However, the material and methods section needs to be re-written to increase the readability and reproducibility.

2) On the other hand, the current version of the manuscript has low reproducibility. Therefore, I highly recommend depositing the codes of the pipeline for each section on GitHub and indicating the URLs of the datasets used in this study. You can also provide the names of the patient/samples used in this analysis as a supplementary table. Especially, for TCGA it is unclear which patients were included in this analysis.

For example: "The samples of LGG patients (n=529) ..."

The official Data Portal reports 516 patients (see the screenshot: https://ibb.co/3Nzd9bw). Is there a reason why you had more patients?
Additionally, n=529 does not match with the numbers of LGG patients shown in Figure A and C.

3) "Finally, we perform the enrichment analysis of PTPRN in LGG."  How did you perform enrichment analysis for a unique gene?

4) "RNA-seq data of TCGA and GTEx in TPM format processed by the Toil pipeline"
Toil (https://github.com/DataBiosphere/toil) can build different pipelines. What is the exact pipeline you used? 
What are the input files you used (raw counts, normalized data, fastq files...etc) and what preprocessing steps did you perform using toil?
Did the authors check if the samples were clustered based on project or any other technical covariates (batch effect)? Please show it with a PCA plot.

5) "P value less than 0.05 was considered significant." Which statistical tests and multiple testing corrections were used?

6) "expression to plotted a Kaplan-Meier survival" -> to plot

7) Please rephrase "The Cancer Genome Atlas provides the TIMER database 10,897 samples across 32 cancer types" accordingly.
Was TIMER developed by the TCGA consortium or only used the public data of TCGA?

8) "2.4. TIMER database to analyze tumor infiltrating immune cells."
Could you please add more details to this section and please make it more clear?

9 ) "PTPRN expression in LGG via different expression modules".
How the PTPRN modules were created?

10) 
- Many times it is written as LLG instead of LGG.
- Please explain what is TME
- "defaulted signature matrix" is this a typo?

11) Why did the authors decide to use 2 different methods to estimate immune cell fractions? What are the main differences between CIBERSORTx and TIMER?

Many thanks  

Reviewer 2 Report

The authors provide a comprehensive bioinformatics analysis of PTPRN expression and prognostic value in LGG. High PTPRN expression correlates with good prognosis and restraints immune infiltration levels (in B cell, CD4+ T cell, macrophage, neutrophil, and dendritic cell) in LGG. GSEA showed that signaling pathways, mainly associated with microenvironment and immune cells, were significantly enriched in the high PTPRN expression group. Albeit, the current study paves the way for more accurate therapeutics for cancer research. I still have some minor suggestions.

1, All figures are highly professional, and the authors should guide the readers to the meaning of the images appropriately; otherwise, it is likely to cause misunderstandings. Therefore, I suggest that the author consider revising these figure legends again.

2,  The authors gave a general answer on gene expression, is there any evidence of different roles in cancer phenotypes of PTPRN family members? Please perform pertinent bioinformatic analyses and provide examples of studies investigating miRNA alteration or DNA methylation (https://biit.cs.ut.ee/methsurv/) in cancer (PMID: 29264942, 34834441, 32560641). 

3, The author demonstrated PTPRN as potential biomarker and correlates with tumor immune infiltration in LGG. Since Connectivity Map (CMap) can be used to discover the mechanism of action of small molecules, functionally annotate genetic variants of disease genes, and inform clinical trials. It would be fascinating if these data could be correlated with other clinical databases. Therefore, I suggest the authors can validate their data via CMap or L1000 platform (PMID: 29195078, 32064155, 35326724).

4, There are a few typo issues for the authors to pay attention. Please unify the writing of scientific terms. “Italic, capital” ? make it consistent throughout the whole manuscript.

5, The font is too small for some of the current figures, and the manuscript also needs English proofreading.

Reviewer 3 Report

Some aspects need to be considered before accepting the manuscript for publication:

-PTPRN is not self-explaining and should be spelled out once in the abstract and the manuscript.

-Your data rely on the correct classification of tutors used for analysis. Please comment on the current WHO CNS classification and discrepancies between such classification and the classification that was employed when TCGA datasets were generated

-Please use a standard of p=0.xxx with three decimals throughout the manuscript

-Oligoastrocytoma was dropped from the 2016 WHO CNS update. However, you provide results including oligoastrocytoma that lack any clinical relevance. Please revise your dataset accordingly.

-The figure quality lacks standards required for publication: examples include  the use of panels instead of simply pasting figures and leaving the reader with the job of figuring out what figure legend addresses which panel instead of adequate labelling. Many figures give p=0. Such statistics is not adequate. You present cumulative survival yet fail to provide a definition in your materials section. Figure 3: "P>0" ?

-No limitations are presented

-You summarise your findings but fail to provide a conclusion

In the end it is an exploratory biostatistical analysis on an old dataset that is not easily translatable to current standards.

Round 2

Reviewer 1 Report

The authors did not attempt to answer all my questions. Here you can find the list of the missing parts which are very essential for the acceptance of this paper (especially points 5,6 and 8).

1) As you also reported, there are duplicated tumor samples for some of the patients. In Supp Table 2 ("1. The Information about LGG Patients ID and Survival Time between Low and High expression groups") there are 528 tumor samples, however, in the Supp Table 1, there are 523 patients.

You have to re-run the survival analysis by removing duplicated samples.

2) "1.The quantification of PTPRN expression and ID in GTEx and LGG patients" reports PTPRN expression per patient. 
For a patient with multiple tumor samples, how did you harmonize the expression values to get a patient-level expression value?

3) "low (n=264) and high (n=263) expression" 
However, Supp Table 2, shows low (n=264) and high (n=264) expression.

4) References 17 and 18 are the same.

5) You have to show how the merged dataset (TCGA+GTEx) looks like using a PCA plot.

6) "...mRNA and protein expression decreased in LGG compared to normal brain tissue in TCGA...".

a) Could you please report the total number of the genes analyzed and the number of the differentially expressed genes (DEGs) obtained in this comparison? 

b) Did you use multiple testing correction for DEGs?

c) Are there any other genes that are highly correlated with PTPRN?

7) Could you please add a reference for "Defaulted signature matrix”? It will help readers to understand what it refers to.

8) The first version of the ms made me feel like you just downloaded the data and then performed the analysis by yourself. Now, after reading the current version, I feel like you performed all the analysis using online tools and merged the results from online resources. I definitely do not have any idea where to put this analysis looking at the methods.

a) Therefore please provide a flow chart or table (as a Supp Material) to explain which parts of the analyses were done using only online tools (Webservers) and which parts were done using in-house pipelines (bash, R, python,...,software,packages, etc).

b) For the in-house pipelines please deposit your codes on GitHub and put its link in the manuscript.

9) Why did the authors decide to use 2 different methods to estimate immune cell fractions? What are the main differences between CIBERSORTx and TIMER? 

a) Sorry but I could not understand your previous response or it did not make perfect sense to me. You added CIBERSORTx to your analysis because you also think that it is better than TIMER? 

b) Just for my better understanding which figures and tables show TIMER and CIBERSORTx results together? 

Many thanks

Reviewer 3 Report

The authors have addressed my reservations in an extensive manner.
